# Cardioprotective Effects of a Nonsteroidal Mineralocorticoid Receptor Blocker, Esaxerenone, in Dahl Salt-Sensitive Hypertensive Rats

**DOI:** 10.3390/ijms22042069

**Published:** 2021-02-19

**Authors:** Asadur Rahman, Tatsuya Sawano, Anupoma Sen, Akram Hossain, Nourin Jahan, Hideki Kobara, Tsutomu Masaki, Shinji Kosaka, Kento Kitada, Daisuke Nakano, Takeshi Imamura, Hiroyuki Ohsaki, Akira Nishiyama

**Affiliations:** 1Department of Pharmacology, Faculty of Medicine, Kagawa University, 1750-1 Ikenobe, Miki-cho, Kita-gun, Kagawa 761-0793, Japan; rahmanma@med.kagawa-u.ac.jp (A.R.); yakuri@med.kagawa-u.ac.jp (A.S.); hossain@med.kagawa-u.ac.jp (A.H.); s19d724@stu.kagawa-u.ac.jp (N.J.); kento-k@med.kagawa-u.ac.jp (K.K.); dnakano@med.kagawa-u.ac.jp (D.N.); 2Division of Pharmacology, Faculty of Medicine, Tottori University, 86 Nishi-cho, Yonago, Tottori 683-8503, Japan; t.sawano@tottori-u.ac.jp (T.S.); timamura@tottori-u.ac.jp (T.I.); 3Department of Gastroenterology and Neurology, Faculty of Medicine, Kagawa University, 1750-1 Ikenobe, Miki-cho, Kita-gun, Kagawa 761-0793, Japan; kobara@med.kagawa-u.ac.jp (H.K.); tmasaki@med.kagawa-u.ac.jp (T.M.); 4Department of Pharmacy, Kagawa University Hospital, 1750-1 Ikenobe, Miki-cho, Kita-gun, Kagawa 761-0793, Japan; kosaka@med.kagawa-u.ac.jp; 5Department of Medical Biophysics, Kobe University Graduate School of Health Sciences, 7-10-2, Tomogaoka, Suma-ku, Kobe, Hyogo 654-0142, Japan; ohsaki@people.kobe-u.ac.jp

**Keywords:** esaxerenone, nonsteroidal mineralocorticoid receptor blocker, cardiac function, salt-sensitive hypertension

## Abstract

We investigated the effects of esaxerenone, a novel, nonsteroidal, and selective mineralocorticoid receptor blocker, on cardiac function in Dahl salt-sensitive (DSS) rats. We provided 6-week-old DSS rats a high-salt diet (HSD, 8% NaCl). Following six weeks of HSD feeding (establishment of cardiac hypertrophy), we divided the animals into the following two groups: HSD or HSD + esaxerenone (0.001%, *w/w*). In survival study, all HSD-fed animals died by 24 weeks of age, whereas the esaxerenone-treated HSD-fed animals showed significantly improved survival. We used the same protocol with a separate set of animals to evaluate the cardiac function by echocardiography after four weeks of treatment. The results showed that HSD-fed animals developed cardiac dysfunction as evidenced by reduced stroke volume, ejection fraction, and cardiac output. Importantly, esaxerenone treatment decreased the worsening of cardiac dysfunction concomitant with a significantly reduced level of systolic blood pressure. In addition, treatment with esaxerenone in HSD-fed DSS rats caused a reduced level of cardiac remodeling as well as fibrosis. Furthermore, inflammation and oxidative stress were significantly reduced. These data indicate that esaxerenone has the potential to mitigate cardiac dysfunction in salt-induced myocardial injury in rats.

## 1. Introduction

The elevated expression of the mineralocorticoid receptor (MR) in the tissues of the failing human heart [1] and in animal models of heart failure [2] suggest a possible role for this receptor in the pathophysiology of cardiovascular diseases. Consequently, MR antagonism has been considered to be an effective therapeutic approach against cardiovascular injury.

MR antagonists have been used clinically for approximately 30 years, and during this time, a series of landmark clinical trials evaluated their efficacy in patients with cardiovascular diseases. The RALES study demonstrated that spironolactone, a steroidal MR antagonist, added on top of an angiotensin-converting enzyme inhibitor in patients with severe heart failure reduces overall morbidity and mortality [3]. Moreover, treatment with eplerenone, another selective steroidal MR antagonist, can reduce cardiovascular events, hospitalization, and death in patients with left ventricular (LV) dysfunction (EPHESUS study) [4] and in patients with systolic heart failure (EMPHASIS HF study) [5]. In these trials, aldosterone levels and sodium status of the patients were normal, suggesting the activation of MR occurs in a ligand (aldosterone) independent manner. Although spironolactone and eplerenone have reportedly been effective in patients with heart failure or cardiovascular diseases, their usage is limited owing to their unfavorable side effects in clinical cases with comorbidities [6].

Salt-dependent hypertension is greatly associated with cardiac hypertrophy in mice [7] and cardiovascular diseases in human subjects [8]. Nevertheless, treatment with steroidal MR antagonists has effectively reduced blood pressure in salt-sensitive hypertensive patients with low circulating aldosterone levels [9,10]. Furthermore, Nagata et al. [11] has demonstrated that treatment with eplerenone attenuated cardiac hypertrophy and failure in Dahl salt-sensitive (DSS) rats fed a diet high in salt. These data suggest that MR antagonism could be a potential therapeutic strategy for patients with salt-sensitive hypertension and subsequent cardiac damage. Although steroidal MR antagonists have been proven to be effective in reducing cardiovascular damage, hyperkalemia still remains an issue during clinical applications. Therefore, there is an unmet need for a novel MR blocker (MRB) with an improved risk-benefit profile for use as an alternative for patients with cardiac injury.

Among the recently developed nonsteroidal MRBs [12,13], esaxerenone has a unique binding mode with MR through the MR-ligand binding domain and large side chains; because of this, esaxerenone has greater affinity and selectivity than other steroidal MR antagonists [14,15]. Several basic and clinical studies have already demonstrated the tolerability and suitability of esaxerenone in patients with essential hypertension [16,17] or in those with hypertension and type 2 diabetes [18,19]. Although an ongoing clinical study [20] aims to assess the impact of esaxerenone in patients with hypertension and heart failure, information on the cardioprotective effects of this medication with salt-dependent hypertension is lacking. High-salt-loaded DSS rats exhibited typical pathophysiology of heart failure, including LV hypertrophy and ventricular dysfunction [11,21]. Therefore, in this study, we evaluate the cardioprotective efficacy of esaxerenone and underlying molecular mechanisms in DSS rats fed a high-salt diet (HSD).

## 2. Results

### 2.1. Effects of Esaxerenone on the Parameters Measured

As shown in Table 1, at 16 weeks of age, HSD-fed DSS rats did not show change in food intake, but showed a significant reduction in body weight as compared with the rats fed a low-salt diet (LSD). Water intake and 24-h urine volume were also significantly higher in the HSD-fed rats than the LSD-fed rats. However, adding esaxerenone with HSD did not affect any of these parameters.

The HSD-fed DSS rats exhibited very low levels of plasma aldosterone concentration (Table 1). In addition, esaxerenone intervention significantly increased plasma aldosterone levels in these rats. Importantly, plasma potassium level was not altered in either the rats fed HSD alone or those concomitantly fed HSD and esaxerenone. Moreover, the HSD-fed rats showed a significant increase in heart and kidney weight without any obvious changes in lung and liver weight. Esaxerenone intervention tended to reduce the value of heart and kidney weight, but these changes were not statistically significant.

### 2.2. Esaxerenone Treatment Improves Survival

All HSD-fed DSS rats died by 24 weeks of age (18 weeks of HSD feeding), whereas 100% of the LSD-fed DSS rats were alive even at 28 weeks, suggesting that the HSD-fed rats exhibited a relatively shorter lifespan (Figure 1A). However, 20% of the HSD-fed rats with concomitant intervention of esaxerenone were alive at 28 weeks. Kaplan-Meier curve analysis of the cumulative probability of survival revealed that esaxerenone treatment significantly improved mean survival time in the HSD-fed DSS rats.

Feeding HSD to animals for 6 weeks gradually and dramatically raised the systolic blood pressure (SBP) (210 ± 4 mmHg) compared with the LSD-fed animals (129 ± 1 mmHg) (Figure 1B). At 12 weeks of age, the HSD-fed DSS rats were further categorized and provided the HSD with and without esaxerenone for the subsequent four weeks. During this period, SBP increased further and reached 236 ± 3 mmHg. However, concomitant treatment with esaxerenone significantly suppressed the further elevation of SBP (214 ± 5 mmHg).

### 2.3. Esaxerenone Treatment Improves Cardiac Function and Remodeling

Echocardiography at 16 weeks of age revealed that interventricular septal thickness both at diastole (IVSd) and systole (IVSs), and LV posterior wall thickness at diastole (LVPWd) significantly increased in the HSD-fed DSS rats as compared with their LSD-fed counterparts (Table 2, Figure 2). However, these parameters were not altered in HSD-fed DSS rats treated with esaxerenone. Furthermore, the HSD-fed rats exhibited systolic dysfunction, with a 25% increase in end-systolic LV diameter (LVIDs) and a reduction in ejection fraction (EF; by 20%), fractional shortening (FS; by 29%), stroke volume (SV, by 23%), and cardiac output (by 23%) (Figure 2A–E), whereas end-diastolic LV diameter (LVIDd) was identical to that of the LSD-fed rats. Remarkably, esaxerenone treatment significantly attenuated systolic dysfucntion by improving all these parameters in the HSD-fed rats.

The mRNA expression of NPPA, NPPB and MYH7 that respectively encode atrial natriuretic peptide (ANP), ventricular or brain natriuretic peptide (BNP), and myosin heavy chain (MHC)-β were significantly up-regulated in the LV tissues of the HSD-fed DSS rats as compared with those of the LSD-fed animals (Figure 2F–H). However, the up-regulation of these cardiac remodeling markers were significantly reduced by esaxerenone treatment in the HSD-fed rats.

### 2.4. Esaxerenone Treatment Reduces Cardiac Fibrosis

Azan-Mallory staining of LV tissue (Figure 3A) indicated that cardiac fibrosis escalated in the interstitial and perivascular regions of myocardium in HSD-fed DSS rats. Strikingly, esaxerenone treatment significantly reduced fibrosis both in the interstitial and perivascular regions. Consistent with Azan-Mallory staining, HSD-fed rats showed a sharp increase in the abundance of mRNA expression of transforming growth factor (TGF)-β, collagen types I and III, and plasminogen activator inhibitor (PAI)-1 (Figure 3B–E). However, esaxerenone treatment in the HSD-fed rats reduced these cardiac fibrotic markers significantly. Moreover, serum and glucocorticoid-regulated kinase (SGK)-1 mRNA expression increased in HSD-fed rats but decreased in esaxerenone-treated rats (Figure 3F).

### 2.5. Esaxerenone Treatment Reduces Cardiac Inflammation

The abundance of mRNA expression of tumor necrosis factor (TNF)-α, interleukin(IL)-6, and CXCL8 were dramatically increased in the HSD-fed DSS rats (Figure 4A–C). Nevertheless, esaxerenone treatment significantly reduced all these inflammatory markers.

### 2.6. Esaxerenone Treatment Reduces Cardiac Oxidative Stress

Cardiac oxidative stress was evaluated by 4-Hydroxynonenal (HNE) immunostaining. As shown in Figure 5A, immunoreactivity against 4-HNE increased in the cytoplasm of cardiomyocytes of LV tissue sections from the HSD-fed rats compared with their LSD-fed counterparts. However, esaxerenone treatment reduced the 4-HNE immunoreactivity. The mRNA expression of genes for the gp47^phox^ and p22^phox^ components of NADPH oxidase were up-regulated in the HSD-fed DSS rats (Figure 5B,C). Treatment with esaxerenone significantly reduced the abundance of the expression of these genes in the HSD-fed DSS rats. Furthermore, LV tissue levels of malondialdehyde (MDA), which is an index of lipid peroxidation, were sharply increased in HSD-fed rats compared with those from LSD-fed DSS rats (Figure 5C). Esaxerenone treatment caused a significant decrease in MDA level in cardiac tissues from DSS rats.

## 3. Discussion

In this study, we showed that treatment with the novel nonsteroidal MRB, esaxerenone significantly prolonged the survival of HSD-fed DSS rats. In addition, esaxerenone treatment attenuated cardiac remodeling and prevented systolic dysfunction. Furthermore, cardiac fibrosis, inflammation, and oxidative stress were reduced after chronic treatment with esaxerenone. These findings suggest that esaxerenone has the potential to ameliorate cardiac dysfunction in salt-loaded DSS rats with a low level of plasma aldosterone.

Steroidal MR antagonists have reduced mortality and hospitalization in patients with heart failure with reduced ejection fraction [3,5]. However, considering that the aldosterone levels were normal in the above mentioned clinical studies, it is difficult to predict the case in salt-dependent cardiac hypertrophy and failure with low aldosterone levels. In the present study, we found that nonsteroidal MRB, esaxerenone significantly enhanced the survival of HSD-fed DSS rats in comparison with their LSD-fed counterparts. Owing to the considerably low plasma aldosterone levels, the observed beneficial effects of MR antagonism could be ligand (aldosterone) independent. A recent study using a transgenic mouse model with conditional cardiomyocyte-specific overexpression of human MR has shown severe arrhythmias and high mortality [22]. In a separate study, the cardiomyocyte-specific overexpression of human MR impairs the nitric oxide-dependent relaxing response in the coronary artery, suggesting a specific role for cardiomyocyte MR in arrhythmia and coronary dysfunction [23]. Therefore, in the present study, prolongation of mean survival time following the antagonizng effects of esaxerenone on cardiac MR could be attributed to the reduction of arrhythmia and coronary dysfunction in the HSD-fed DSS rats.

Salt-sensitive hypertension is an independent risk factor for cardiovascular disease and mortality [9,24]. In the prsent study, high-salt loading significantly increased the SBP in DSS rats; however, esaxerenone treatment abolished the gradual increase in BP, which might be associated with the cardioprotective effects of esaxerenone. Previous studies in cardiomyocyte-specific MR knockout (KO) mice that underwent transverse aortic constriction showed improvements in LV dysfunction [25]. Consistently, cardiac remodeling was attenuated in cardiomyocyte-specific MR-KO mice following myocradial infarction [26]. In these mice, cardiac remodeling and functions improved; however, cardiac hypertrophy remained unchanged. Consistent with previous reports [11], we found that the HSD-fed DSS rats developed severe LV hypertrophy, evident by a significant increase in LV weight/tibial length. Furthermore, cardiac remodeling was significantly altered, which was associated with increases in NPPA, NPPB and MYH7 mRNA levels. Moreover, the HSD-fed rats exhibited ventricular systolic dysfucntion with an increase in LVIDs and a reduction in EF, FS, SV and cardiac output. Notably, esaxerenone treatment significantly improved cardiac remodeling and ventricular systolic dysfunction without any changes in ventricular hypertrophy. These data suggest that cardiac MR antagonism with nonsteroidal MRB is cardioprotective in salt-dependent cardiac dysfunction in DSS rats.

Cardiac fibrosis is often observed in association with hypertension, and cardiac hypertrophy and failure. These changes contribute to myocardial and myocyte stiffening, and hence, can impair ventricular function. In this study, HSD-fed DSS rats showed an increase in the extent of interstitial and perivascular fibrosis in LV tissues. Treatment with esaxerenone drastically reduced fibrosis in these rats, which is consistent with a previous report on the nonsteroidal antagonist, eplerenone [11]. TGF-β is a pleiotropic mediator that is critical in cardiac fibrosis [27]. Furthermore, TGF-β plays an important role in the transformation from fibroblasts to myofibroblasts and promotes extracellular matrix deposition [28]. In the present study, treatment with esaxerenone significantly reduced mRNA experssion in the HSD-fed DSS rats. In addition, collagens are considered the major extracellllular matrix protein in the heart, and among different collagen isoforms, collagen types I and III constitute approximately 85% of the cardiac interstitium [29]. Importantly, the abundance of both collagen types I and III was greatly increased in HSD-fed rats; however, their reduction was consequently facilitated by esaxerenone treatment in these animals. Furthermore, studies have shown that incresead expression of PAI-1 in cardiomyocyte plays a critical role in cardiac fibrosis through the regioal induction of cytokines and subsequent progression of LV dysfunction [30]. We also found that PAI-1 mRNA was significantly increased in the HSD-fed DSS rats, and treatemnt with esaxerenone significantly reduced the level of PAI-1 mRNA expression. Furthermore, activation of the renin–angiotensin system plays an important pathophysiological role in hypertensive cardiac fibrosis and remodeling. Recent studies have demonstrated that SGK-1 plays a critical role in angiotensin or mineralocorticoid-induced cardiac fibrosis [31,32]. In the present study, we found that SGK-1 mRNA expression increased in HSD-fed DSS rats. However, treatment with esaxerenone significantly reduced SGK-1 mRNA levels, which could be associated with the reduction of cardiac fibrosis. Collectively, all the data suggest that esaxerenone treatment attenuates cardiac fibrosis by down-regulating most of the factors involved in this process.

Cardiomyocyte-specific MR-KO mice have shown less inflammatory cell infiltration and fibrosis induced by deoxycorticosterone and salt [33]. Moreover, TNF-α KO mice have exhibited worst LV remodeling through cardiac inflammatory or fibrogenic responses [34]. In addition, CXCL8 is a key regulator in the influx of inflammatory cells in inflammatory processes [35]. In the present study, the mRNA abundances of TNF-α, IL-6 and CXCL-8 were significantly up-regulated in HSD-fed DSS rats. In contrast, treatment with esaxerenone significantly down-regulated all these genes related to cardiac inflammation, suggesting a benefial role of MR antagonization with a nonsteroidal MRB.

Oxidative stress activates MR in cardiomyocytes via the ligand-independent Rac1-dependent pathway [36]. Moreover, heterozygous deletion of Rac1 in cardiomyocytes has revealed that oxidative stress-stimulated Rac1 plays a role in myocardial dysfunction through MR activation [37]. Recently, researchers also have demonstarted that oxidative stress can induce motochondrial DNA damage leading to the catastrophic cylce of mitochondrial functional decline and cardiomyocyte injury in the failing heart [38]. In the present study, we assesed the oxidative stress by immunostaining with 4-HNE, a byproduct of lipid peroxidation. The LV tissue from HSD-fed DSS rats exhibited a strong immunoreactivity for 4-HNE, whereas a faint immunostaining was detected in animals with esaxerenone treatment. Moreover, the mRNA abundances of genes for the components of NADPH oxidase gp47^phox^ and p22^phox^ significantly increased in HSD-fed DSS rats, whereas treatment with esaxerenone significantly down-regulated the expression of these genes. Moreover, lipid peroxidation, which was evaluated using MDA levels, was greatly increased in HSD-fed DSS rats, but it decreased in esaxerenone-treated DSS rats. These findings are consistent with the results for esaxerenone, which elicits cardioprotective effects through the suppression of oxidative stress, and this is similar to a previous report on eplerenone [11].

In the present study, the survival data indicated a beneficial effect of esaxerenone on prolonging the life span of salt-loaded DSS rats; however, a limitation of this study is that a relatively small number of animals was used for the survival analysis. Moreover, we failed to measure the ventricular diastolic function due to the limitations of our machine performance. Nevertheless, another nonsteroidal MRB, finerenone has been shown to be effective in attenuating diastolic dysfunction in chronic kidney diseases mice [39], transgenic mice with cardiac-specific overexpression of Rac1 [40], and deoxycorticosterone acetate-/salt-challenged rats [41]. These data suggest that esaxerenone might be competent in preventing ventricular diastolic dysfunction. Although our data clearly indicated that the improvement of cardiac dysfunction was associated with a reduction in cardiac fibrosis, inflammation, and oxidative stress, future studies should be undertaken to determine the beneficial effects of nonsteroidal MR blockers on cardiac cell death especially apoptosis, necrosis, or necroptosis. Importantly, esaxerenone is currently under investigation in a clinical study focusing on LV diastolic function in patients with hypertension and heart failure [20].

In summary, we observed that HSD-fed DSS rats exhibited ventricular cardiac dysfunction accompanied with a dramatic increase in SBP. This dysfunction is associated with cardiac fibrosis, inflammation, and oxidative stress. However, treatment with esaxerenone effectively reduced cardiac dysfunction as well as its associated pathological features in HSD-fed DSS rats.

## 4. Materials and Methods

### 4.1. Ethical Approval

Experimental protocols (Protocol No. 18627) were approved by the Animal Experimentation Ethics Committee at Kagawa University. All experimental procedures were conducted to conform to the guidelines of the care and use of animals established by Kagawa University.

### 4.2. Animals

DSS rats were preferred as a model for cardiac hypertrophy. Five-week-old male DSS-Iwai rats (Japan SLC, Inc., Shizuoka, Japan) were maintained in specific pathogen-free facilities under controlled temperature (24 ± 2 °C) and humidity (55 ± 5%) conditions with a 12-h light–dark cycle. Rats were given standard chow (0.5% NaCl) for 1 week. The experiments were conducted in two phases with different sets of animals to determine the survival and cardiac function. At 6 weeks of age, rats weighting 160–180 g were divided (based on the SBP) into either a LSD (0.3% NaCl, *n* = 10 for survival, and *n* = 5 for cardiac function study) or a HSD (8% NaCl, *n* = 20 for survival, and *n* = 20 animals for cardiac function study) group, and they were treated for 6 weeks. At 12 weeks of age, HSD-fed rats were further subdivided into the following two groups: HSD continued (*n* = 10) or 0.001% esaxerenone (*w/w*) was added to the HSD (*n* = 10). The concentration of esaxerenone in the HSD was calculated on the basis of our previous report [42], where DSS rats were gavaged daily with esaxerenone at a dose of 1 mg/kg body weight. In the survival study, all HSD-fed animals died by 24 weeks of age. However, we continued the experiment until 28 weeks of age (16 weeks of intervention), at which point only two animals in the esaxerenone treatment group were still alive. In the cardiac function study, we administered the intervention for 4 weeks and continued the experiments until 16 weeks of age. Esaxerenone was provided by Daiichi-Sankyo Co., Ltd. (Tokyo, Japan).

### 4.3. Blood Pressure

SBP was measured in conscious rats by tail-cuff plethysmography (BP-98A; Softron Co., Tokyo, Japan). Conscious rats were placed in a plastic holder resting on a warm pad at 37 °C. After 15 min of acclimatization, SBP was measured at least five consecutive times, and the three later values were averaged for each rat.

### 4.4. Echocardiography

Echocardiographic measurements were performed at 16 weeks of age. Rats were anesthetized with inhalation of 2% isoflurane and maintained with 1.5% isoflurane. Transthoracic echocardiography was performed using a LOGIQ e system (GE Healthcare Japan Co., Tokyo, Japan) equipped with a 22-MHz linear transducer. Parasternal long-axis M-mode tracings of the LV were recorded to measure the LVIDs, LVIDd and to assess the cardiac function such as FS, EF, SV and cardiac output. These parameters were averaged based on three measurements from each animal.

### 4.5. Sample Collection

At the end of the experiment, the animals were anesthetized with isoflurane, and blood was immediately collected from the abdominal aorta to the Ethylenediaminetetraacetic acid ontaining tubes on ice. Whole blood was centrifuged at 4 °C for 10 min to separate the plasma. Subsequently, animals were euthanized by an intraperitoneal overdose of pentobarbital (250 mg/kg), and the heart tissues were harvested. After measuring the heart weight, the isolated LV tissues were cut into pieces and fixed in a 10% buffered paraformaldehyde. The remaining tissues were kept in tubes with RNAlater or snap frozen in liquid nitrogen.

### 4.6. Histopathological Examination

Formalin-fixed cardiac LV tissues were embedded in paraffin, cut into 4-μm sections, and mounted on slides. The sections were then stained with Mallory-Azan reagent. Images were captured by a light microscope (BX-51/DP-72; Olympus, Tokyo, Japan).

### 4.7. Immunohistochemistry

After treatment with an endogenous peroxidase activity-blocking solution (0.3% hydrogen peroxide in methanol) for 30 min at room temperature (RT; 22–25 °C), sections were incubated with mouse monoclonal antibody against 4-HNE (HNEJ-2) (1:20, JaICA, Shizuoka, Japan) in a humidified chamber overnight at RT. Then, sections were incubated with Histofine Simple Stain Rat MAX-PO (Nichirei Bioscience, Tokyo, Japan) for 30 min at RT. Subsequently, sections were treated with diaminobenzidine (Nichirei Bioscience) and counterstained with Mayer’s hematoxylin.

### 4.8. Real-Time Reverse Transcriptase PCR

RNA was isolated from LV tissues by the phenol–chloroform extraction method and cDNA prepared as described previously [38]. NPPA, NPPB, MYH7, TGF-β, collagen types I and III, PAI-1, TNF-α, IL-6, CXCL8, gp47^phox^, p22^phox^, and SGK-1 mRNA expression were analyzed by real-time PCR using an ABI Prism 7000 system with Power SYBR Green PCR Master Mix (Applied Biosystems, Foster City, CA, USA). The list of primers is shown in Appendix A. All data showed the relative differences between the LSD-fed DSS rats and the other groups after normalization to mRNA expression of the 18s gene.

### 4.9. Plasma Electrolytes and Aldosterone

Plasma potassium was measured by an automated analyzer (7020-Automatic Analyzer; Hitachi High-Technologies, Tokyo, Japan). Plasma aldosterone concentration was measured using a competitive radioimmunoassay (RIA) as previously described [42]. In this RIA, aldosterone from the sample competes with the aldosterone that is labeled with iodine 125 (tracer) for the specific sites of the antiserum that is coated onto the tubes. The degree of binding is inversely proportional to the aldosterone concentration of the sample.

### 4.10. Cardiac TISSUE Level of MDA

LV tissue homogenates were used to determine the MDA levels using the thiobarbituric acid reactive substances method in accordance with the protocol that was provided with the assay kit (Elabscience, Houston, TX, USA). Protein concentration in the tissue homogenates was measured by the Bradford assay, and MDA levels were normalized to the homogenate’s total protein content. The data are presented as nmol/mg protein.

### 4.11. Statistical Analysis

Data are presented as the means ± SEM. We used one-way analysis of variance (ANOVA) followed by the Newman-Keuls multiple-comparison test for all cross-sectional, one-factor data to compare values in the LSD-fed DSS rats with those treated with HSD alone or with concomitant esaxerenone treatment. The longitudinal data (SBP) were analyzed by two-way ANOVA followed by the Bonferroni post hoc test. A value of *p* < 0.05 was considered statistically significant.

## Figures and Tables

**Figure 1 ijms-22-02069-f001:**
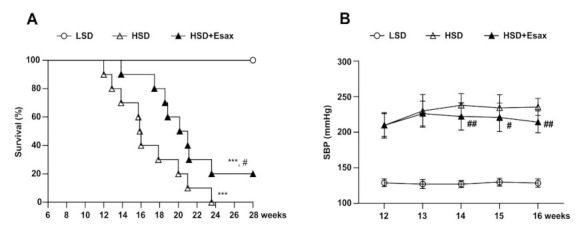
Esaxerenone treatment increases survival and reduces SPB in HSD-fed DSS rats. (**A**) Kaplan–Meier survival analyses in the LSD (*n* = 10), HSD (*n* = 10), and HSD + Esax (*n* = 10) groups of DSS rats. The HSD-fed rats exhibited decreased survival as compared with the LSD-fed rats. The HSD-fed rats receiving esaxerenone showed significantly higher survival than did the HSD-fed rats. (**B**) Time-dependent changes of SBP during feeding LSD, HSD, and HSD + Esax. SBP was gradually increased in the HSD-fed rats; however, esaxerenone treatment significantly reduced SBP. *** *p* < 0.001 vs. LSD; ^#^
*p* < 0.05, ^##^
*p* < 0.01 vs. HSD.

**Figure 2 ijms-22-02069-f002:**
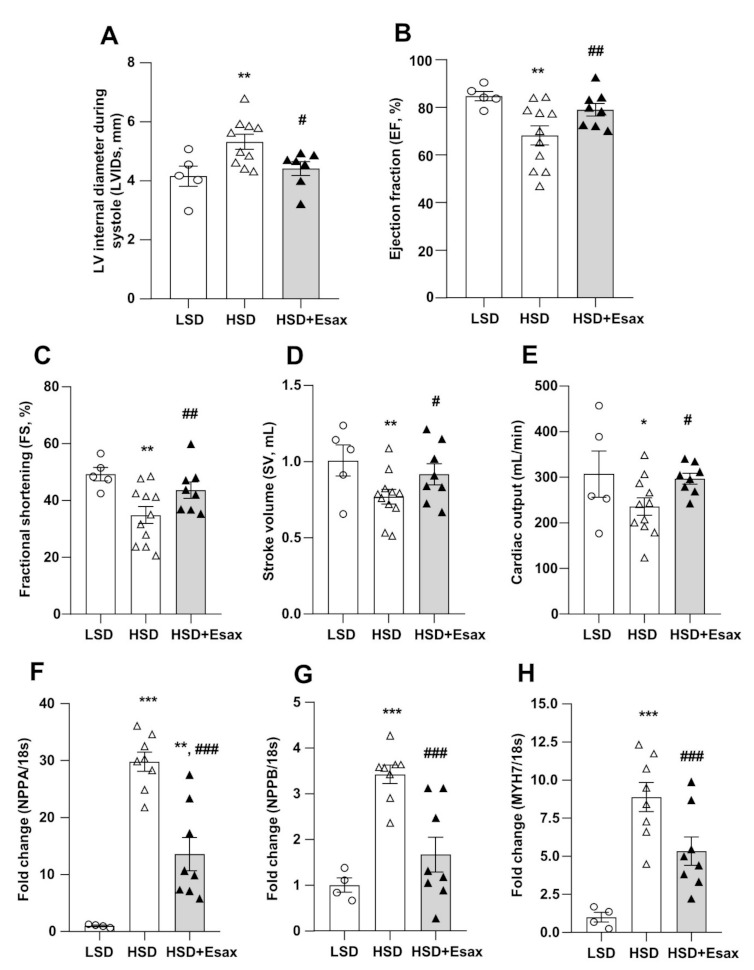
Esaxerenone treatment improves cardiac function and remodeling in HSD-fed DSS rats. The cardiac morphology and function of the 16-week-old rats were assessed by echocardiography. (**A**) LV internal diameter during systole (LVIDs), (**B**) EF, (**C**) FS, (**D**) SV, and (**E**) cardiac output. The HSD-fed rats showed worsening cardiac function; however, esaxerenone significantly improved the cardiac dysfunction. The mRNA expression of cardiac remodeling markers: (**F**) NPPA, encode atrial natriuretic peptide, ANP; (**G**) NPPB, encode brain natriuretic peptide, BNP; and (**H**) MYH7, encode myosin heavy chain (MHC)-β. * *p* < 0.05, ** *p* < 0.01, *** *p* < 0.001 vs. LSD; ^#^
*p* < 0.05, ^##^
*p* < 0.01, ^###^
*p* < 0.001 vs. HSD.

**Figure 3 ijms-22-02069-f003:**
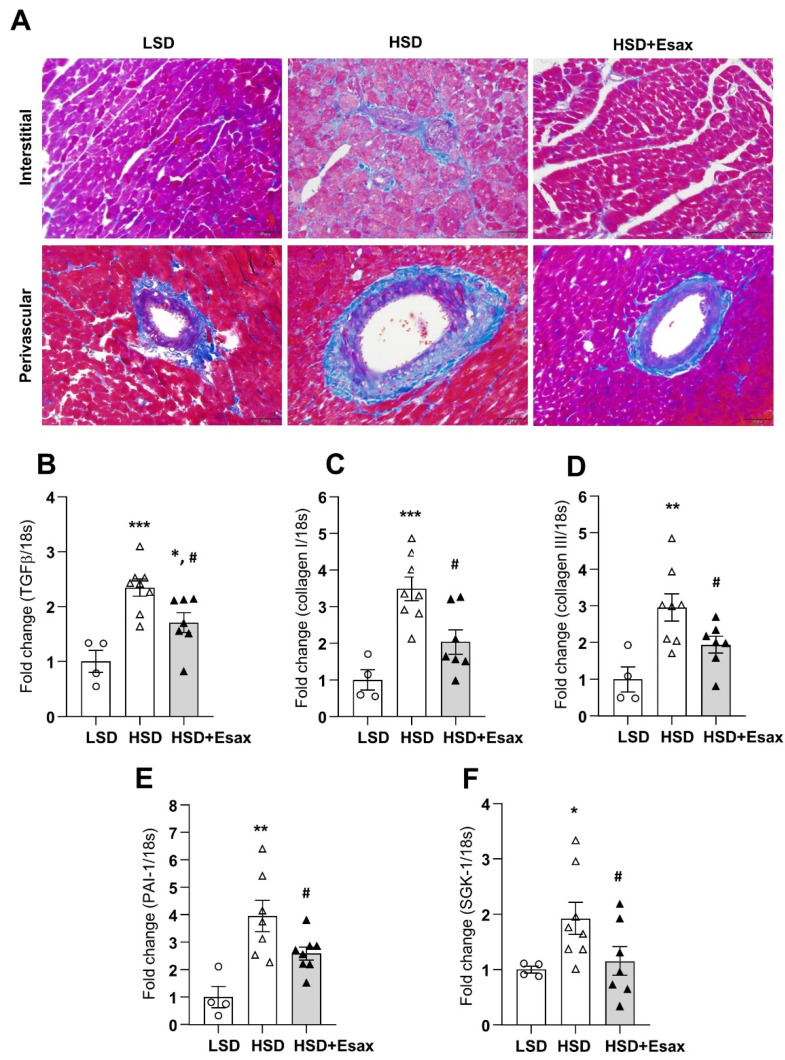
Esaxerenone treatment improves cardiac fibrosis in HSD-fed DSS rats. (**A**) Representative Azan staining in interstitial and perivascular regions of LV tissues, magnification 200×, scale bar: 50 μm. Obvious increase in fibrosis in the HSD-fed rats, which was attenuated in esaxerenone treatment group. The mRNA expression of cardiac fibrotic markers: (**B**) transforming growth factor (TGF)-β; (**C**) collagen type I; (**D**) collagen type III; (**E**) plasminogen activator inhibitor (PAI)-1; and (**F**) serum and glucocorticoid-regulated kinase (SGK)-1 in the LV tissue obtained from the LSD, HSD, and HSD + Esax treatment groups. * *p* < 0.05, ** *p* < 0.01, *** *p* < 0.001 vs. LSD; ^#^
*p* < 0.05 vs. HSD.

**Figure 4 ijms-22-02069-f004:**
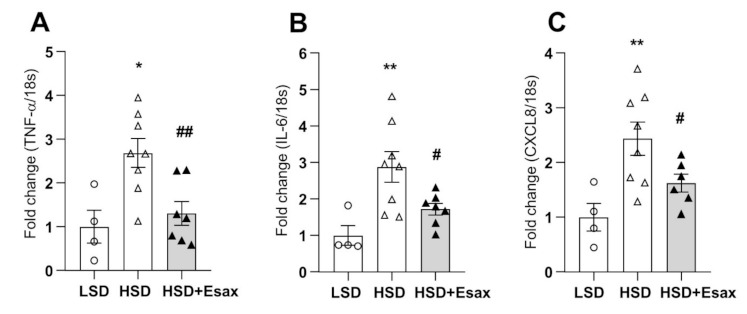
Esaxerenone treatment attenuates cardiac inflammation in HSD-fed DSS rats. The mRNA expression of inflammatory markers: (**A**) tumor necrosis factor (TNF)-α; (**B**) interleukin (IL)-6; and (**C**) CXCL8. The inflammatory markers were escalated in the HSD-fed rats; however, esaxerenone treatment dramatically reduced all these markers. * *p* < 0.05, ** *p* < 0.01 vs. LSD; ^#^
*p* < 0.05, ^##^
*p* < 0.01 vs. HSD.

**Figure 5 ijms-22-02069-f005:**
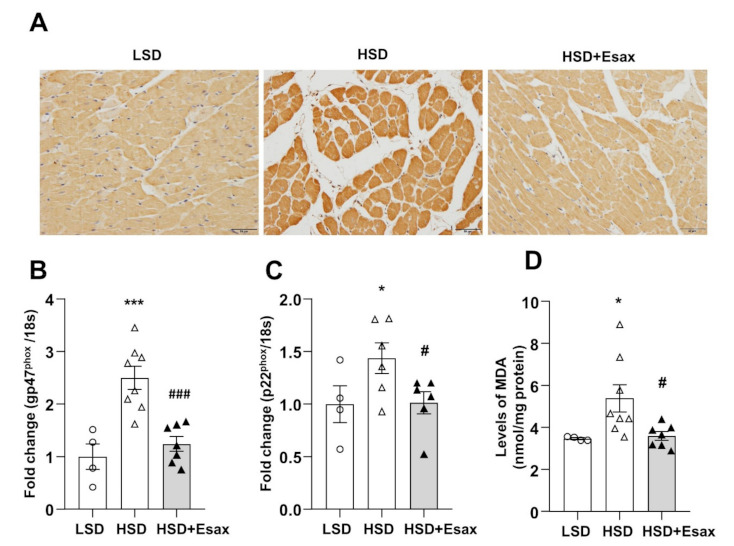
Esaxerenone treatment suppresses oxidative stress in HSD-fed DSS rats. (**A**) Immunohistochemistry of 4-hydroxynonenal (HNE), magnification 200×, scale bar: 50 μm. The mRNA expression of oxidative stress markers: (**B**) gp47^phox^ and (**C**) p22^phox^. (**D**) LV tissue content of malondialdehyde (MDA). These oxidative stress markers were up-regulated in the HSD-fed DSS rats but down-regulated in those receiving the esaxerenone treatment. * *p* < 0.05, *** *p* < 0.001 vs. LSD; ^#^
*p* < 0.05, ^###^
*p* < 0.001 vs. HSD.

**Table 1 ijms-22-02069-t001:** Effects of esaxerenone on the parameters *measured* in DSS rats at 16 weeks of age.

Parameters	LSD	HSD	HSD + Esax
Body weight (g)	390 ± 5	325 ± 10 ***	334 ± 8 ***
Food intake (g)	15.7 ± 0.7	16.2 ± 1.2	17.5 ± 1.4
Water intake (mL)	21.9 ± 0.9	82.9 ± 4.8 ***	84.9 ± 6.8 ***
Urine volume (mL/24 h)	6.2 ± 0.7	66.7± 5.9 ***	64.2 ± 5.5 ***
Plasma aldosterone concentration (pg/mL)	134 ± 15	70 ± 14	146 ± 29 ^#^
Plasma potassium (mmol/liter)	5.6 ± 0.3	5.8 ± 0.3	5.5 ± 0.6
Heart weight/body weight (mg/gm)	3.1 ± 0.1	5.1 ± 0.2 ***	4.7 ± 0.2 ***
Heart weight/tibial length (mg/mm)	30.4 ± 0.8	40.3 ± 1.1 ***	39.3 ± 1.2 ***
LV weight/body weight (mg/gm)	2.3 ± 0.1	3.8 ± 0.1 ***	3.5 ± 0.1 ***
LV weight/tibial length (mg/mm)	22.4 ± 0.7	30.7 ± 0.7 ***	29.3 ± 0.7 ***
Left kidney/tibial length (mg/mm)	35.7 ± 1.6	47.7 ± 2.0 ***	40.5 ± 4.5 *
Right kidney/tibial length (mg/mm)	34.8 ± 1.8	48.0 ± 2.1 ***	45.9 ± 0.8 **
Lung weight/tibial length (mg/mm)	46.7 ± 4.0	46.7 ± 3.8	46.1 ± 1.8
Liver weight/tibial length (mg/mm)	338 ± 12	317 ± 16	307 ± 10

LSD, low-salt diet; HSD, high-salt diet; Esax, esaxerenone; LV, left ventricle. Values are means ± SEM. * *p* < 0.05, ** *p* < 0.01, *** *p* < 0.001 vs. LSD; ^#^
*p* < 0.05 vs. HSD.

**Table 2 ijms-22-02069-t002:** Echocardiographic data at 16 weeks of age.

Parameters	LSD	HSD	HSD + Esax
IVSd (mm)	1.9 ± 0.1	2.8 ± 0.1 ***	2.6 ± 0.1 **
IVSs (mm)	3.3 ± 0.1	3.8 ± 0.1 *	3.9 ± 0.1 *
LVIDd (mm)	8.1 ± 0.4	7.9 ± 0.1	8.1 ± 0.2
LVPWd (mm)	2.0 ± 0.1	2.7 ± 0.1 ***	2.5 ± 0.1 **
LVPWs (mm)	3.4 ± 0.1	3.6 ± 0.1	3.7 ± 0.1
EDV (mL)	1.2 ± 0.1	1.1 ± 0.1	1.2 ± 0.1
ESV (mL)	0.19 ± 0.04	0.36 ± 0.05 **	0.24 ± 0.03 ^#^
HR (beats/min)	311 ± 43	306 ± 14	332 ± 19

IVSd, inter ventricular septal thickness at diastole; IVSs, inter ventricular septal thickness at systole; LVIDd, left ventricular internal diameter at end diastole; LVPWd, left ventricular posterior wall thickness at diastole; LVPWs, left ventricular posterior wall thickness at systole; EDV, end-diastolic volume; ESV, end-systolic volume; HR, heart rate. Values are means ± SEM. * *p* < 0.05, ** *p* < 0.01, *** *p* < 0.001 vs. LSD; ^#^
*p* < 0.05 vs. HSD.

## Data Availability

The data presented in this study are available in this article or Appendix A.

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
