# Peer review of "Cardioprotective Effects of a Nonsteroidal Mineralocorticoid Receptor Blocker, Esaxerenone, in Dahl Salt-Sensitive Hypertensive Rats"

_ijms, 2021, doi:10.3390/ijms22042069_

Round 1
Reviewer 1 Report
In their revised manuscript entitled: “Cardioprotective effects of a nonsteroidal mineralocorticoid receptor blocker, esaxerenone, in Dahl salt-sensitive hypertensive rats”, Asadur Rahman et al. have made several changes, according to the concerns raised by the reviewer and have ameliorated the quality and scientific merit of the data.
However, several principal concerns have not been dealt with.
In particular, the originality and novelty of the present study still consist an issue given the numerous papers already published in this specific field.
Another major concern is the fact that the results presented are based on data acquired from 2 rats that have survived, since the rest of the HSD-fed rats with esaxerenone died and there were no “survivors” in the HSD-fed rats group. This is a major issue that cannot be overlooked! This is the reason the reviewer had asked for the study to be repeated starting with a much higher number of animals, since the authors intended to further divide them as the study progressed….
Last but not least, taking into consideration the mortality levels in the HSD-fed group, as well as the limited survival in the other experimental groups, the study would actually benefit a lot from the investigation of the death mechanisms induced, that mediate the observed responses (apoptosis, necrosis, necroptosis or other).
Finally, with this MS aiming to be published in a special issue focusing on the Renin-angiotensin-aldosterone system (RAAS), an effort should be made to evaluate any potential involvement of this system in the responses under investigation.
Author Response
Responses to the comments raised by Reviewer #1
(General comment)
This study provides evidence on positive effects of esaxerenone treatment on survival in an animal model of salt-induced myocardial injury. The authors used several experimental approaches to identify the mechanisms involved. There are only few points which need clarification or correction.
Response
We thank the reviewer for his/her valuable comments, which have helped us to improve the manuscript. In response to the concerns and issues that were raised, we have made extensive changes to the manuscript, which are detailed below in a point-by-point manner.
Major Comments
Comment-1
There is a typing error in line 49. Replace “…clinical trials have undertaken…” by “…clinical trials have been undertaken…”
Response
We apologize for our error. We have revised the text on page 2, lines 48–50, as follows:
“MR antagonists have been used clinically for approximately 30 years, and during this time, a series of landmark clinical trials have evaluated their efficacy in patients with cardiovascular diseases.”
Comment-2
Lines 60-61 – Please, specify whether you are mentioning human or animal studies.
Response
We have revised the text on page 2, lines 60-61, as follows:
“Salt-dependent hypertension is greatly associated with cardiac hypertrophy in mice [7] and cardiovascular diseases in human subjects [8].”
Comment-3
Please, rename the subtitle 2.1 and the title of Table 1 without the use of words “physiological parameters”. Physiological means healthy functioning. Most parameters can have physiological or pathological values.
Response
We revised subtitle 2.1 as follows: “Effects of esaxerenone on various parameters”.
We also revised the title of “Table 1” as follows: “Effects of esaxerenone on various parameters in DSS rats at 16 weeks of age”.
Comment-4
Methods, around line 299 – It should be described how and why the dose of esaxerenone was selected. The value 0.001% in the diet is not informative.
Response
In our previous study (Li et al. Hypertens. Res. 2019, 42, 769–778), we treated the DSS rats with esaxerenone (1 mg/kg body weight) daily by oral gavage, which is equivalent to 0.2 mg/rat when the average body weight is approximately 200 g. Because the dose in the previous experiment was effective, we choose the same dose of esaxerenone, but administered it with the chow in the present study. DSS rats weighing approximately 200 g usually eat approximately 20 g of chow daily. Therefore, we added 0.001% esaxerenone (w/w) with 8% NaCl chow (high salt diet, HSD), which is equivalent to an approximate intake of 0.2 mg of esaxerenone for each rat. To improve the manuscript’s clarity, we revised the text, as follows:
Materials and methods (page 10, line 314-page 11, line 318)
“At 12 weeks of age, HSD-fed rats were further subdivided into the following two groups: HSD continued (n = 10) or 0.001% esaxerenone (w/w) was added to the HSD (n = 10). The concentration of esaxerenone in the HSD was calculated on the basis of our previous report [42], where DSS rats were gavaged daily with esaxerenone at a dose of 1 mg/kg body weight.”
Comment-5
Line 348 – The basic characteristics of the aldosterone radioimmunoassay should be provided.
Response
We explained the basic characteristics of aldosterone radioimmunoassay, and thus, the text was revised as follows:
Materials and methods (page 12, lines 366-370)
“Plasma aldosterone concentration was measured using a competitive radioimmunoassay (RIA) as previously described [38]. In this RIA, aldosterone from the sample competes with aldosterone that is labeled with iodine-125 (tracer) for the specific sites of the antiserum that is coated onto the tubes. The degree of binding is inversely proportional to the aldosterone concentration of the sample.”
Comment-6
Lines 199-201 – The first sentence of the Discussion is not needed; it should better be deleted.
Response
We have removed the sentence from the Discussion section.
Comment-7
In the Discussion, I miss a comparison of the effects of esaxerenone with the effects of steroid antagonists or finerenone in similar animal models, if the data is available.
Response
Cardioprotective effects of finerenone have been reported in deoxycorticosterone acetate-/salt-challenged rats [Kolkhof et al. J Cardiovasc Pharmacol. 2014; 64(1):69-78] and transgenic mice with cardiac-specific overexpression of Rac1 [Lavall et al. Biochem Pharmacol. 2019; 168:173-183]. Additionally, the steroidal mineralocorticoid antagonist, eplerenone, was shown to be effective in reducing the left ventricular hypertrophy and fibrosis in the same animal model (DSS rats) [Nagata et al. Hypertension. 2006; 47(4):656-64]. Although we mentioned the information on eplerenone in DSS rats in the Introduction (page 2, lines 63-65), we have now also included all this information in the Discussion section, as follows:
Discussion (page 9, lines 245-248)
“In this study, HSD-fed DSS rats showed an increase in the extent of interstitial and perivascular fibrosis in LV tissues. Treatment with esaxerenone drastically reduced fibrosis in these rats, which is consistent with a previous report on the nonsteroidal antagonist, eplerenone [11].”
Discussion (page 10, lines 287-289)
“These findings are consistent with the results for esaxerenone, which elicits cardioprotective effects through the suppression of oxidative stress, and this is similar to a previous report on eplerenone [11].”
Discussion (page 10, lines 291-294)
“However, another nonsteroidal MRB, finerenone, has been shown to be effective in attenuating diastolic dysfunction in CKD mice [39], transgenic mice with cardiac-specific overexpression of Rac1 [40], and deoxycorticosterone acetate-/salt-challenged rats [41].”
Reviewer 2 Report
The authors have sufficiently considered all my comments.
The corrected term "on various parameters" should better be replaced by "on the parameters measured" in the subtitle 2.1 and in the legend to Table 1. This may be performed by the editorial office.
Author Response
Responses to the comments raised by Reviewer #2
(General comment)
In their manuscript entitled: “Cardioprotective effects of a nonsteroidal mineralocorticoid receptor blocker, esaxerenone, in Dahl salt-sensitive hypertensive rats”, Asadur Rahman et al. report their results showing the beneficial effects of esaxerenone in Dahl salt-sensitive hypertensive rats.
Response
We thank the reviewer for his/her valuable comments, which have helped us to improve the manuscript. In accordance with the concerns and issues that were identified, we performed additional experiments and made extensive changes to the text of the manuscript, which are described below in a point-by-point manner.
Major Comments
Comment-1
Given that as the authors themselves report in the Introduction: “Several basic and clinical studies have already demonstrated the tolerability and suitability of esaxerenone in patients with essential hypertension [16,17] or in those with hypertension and type 2 diabetes [18,19]. Although an ongoing clinical study [20] aims to assess the impact of esaxerenone in patients with hypertension and heart failure….” both the novelty and scientific significance of this study consist an issue.
Response
Thank you for identifying this issue. To date, all basic and clinical studies that have been conducted mostly focused on the antihypertensive and/or renoprotective effects of esaxerenone. The ongoing clinical study concentrates on the impact of esaxerenone on left ventricular dysfunction in heart failure patients with hypertension. The aim of this study was to explore the molecular mechanism of the cardioprotective effects of esaxerenone in salt-sensitive hypertensive rats, which are known to exhibit typical pathophysiology of heart failure including left ventricular hypertrophy and ventricular dysfunction. We believe the data that we obtained in the present study will provide important insights for clinicians. To provide the readers with an accurate understanding of this aim to investigate the cardioprotective effect of esaxerenone, these issues have been clearly described in the Introduction section (page 2, lines 73-81).
Comment-2
Another major concern is the experimental design of the study. In particular, as the authors state: “HSD-fed DSS rats died by 24 weeks of age (18 weeks of HSD feeding), whereas 100% of the LSD-fed DSS rats were alive even at 28 weeks, suggesting that the HSD-fed rats exhibited a relatively shorter lifespan (Figure 1A). However, 20% of the HSD-fed rats with concomitant intervention of esaxerenone were alive at 28 weeks”. Looking into the materials and methods section one realizes that in fact the 20% corresponds to the 2 rats that survived, out of the 10 that originally belonged to this group. This cannot be accepted as statistically significant. The one group has null survival and the other one, that is supposed to prove the beneficial effect of esaxerenone, contains only 2 rats. How can this be reliable? Unfortunately, this study has to be repeated starting with a much higher number of animals, since the authors intended to further divide them as the study progressed.
Response
We thank the reviewer for pointing out this issue. We agree with the reviewer’s opinion that survival studies should be performed with a large number of animals. This issue was pointed out to us in advance by the person who was in charge when we applied for approval of the animal protocols. In Japan, from the perspective of animal welfare, we have been advised to reduce the numbers of animals used as much as possible. Our previous studies have shown that HSD-fed DSS rats are often in very poor general condition due to worsening heart failure or other reasons before they die. In accordance with these experts’ suggestions, we decided to minimize the number of animals that were used in the survival studies.
A Kaplan–Meier curve analyzes the cumulative probability of survival over a given time using the log-rank test. The null hypothesis for a log-rank test is that all the groups with a different intervention have the same survival. The log-rank test determines if the observed number of events (death in the present study) in each group is significantly different from the expected number. Thus, the Kaplan–Meier survival analysis considers all the events or death time points during the experimental period rather than only the last time point. We have explained that 20% of the animals were alive in the esaxerenone treatment group at 28 weeks of age, but there was a significant difference in the overall survival throughout the experiment.
To improve clarity in the manuscript, we have revised the text, as follows:
Results (page 3, lines 103-105)
“Kaplan-Meier curve analysis of the cumulative probability of survival revealed that esaxerenone treatment significantly improved mean survival time in the HSD-fed DSS rats.”
Comment-3
Another concern that confuses the reader is the time points that the echocardiographic data and the tissue samples were collected. Since the HSD-fed DSS rats all died, the authors collected data from diverse time points that vary and are not consistent, so that one can extract comparable conclusions between the various experimental groups.
Response
We apologize for any confusion that was caused by our explanation of the experimental procedure in the Methods section. We conducted the experiments in two phases with two different sets of animals. In the first phase, we recorded only the death and survival of animals in the different groups, and we did not collect any samples when the animals died at different time points throughout the experiment. In this survival study, all animals in HSD-fed groups died within 24 weeks, and we continued the experiments until 28 weeks of age. In the second phase, we used almost the same number of animals with the same intervention as in the first phase. However, in this phase, we continued the experiment until 16 weeks of age, performed echocardiography, and then sacrificed all the rats to collect the plasma and tissue samples for molecular investigation. To improve clarity in the manuscript, we have revised the text, as follows:
Materials and methods (page 10, line 311-page 11, line 323)
“The experiments were conducted in two phases with different sets of animals to determine the survival and cardiac function. At 6 weeks of age, rats weighting 160–180 g were divided (based on the SBP) into either a LSD (0.3% NaCl, n = 10 for survival, and n = 5 for cardiac function study) or a HSD (8% NaCl, n = 20 for survival, and n = 20 animals for cardiac function study) group, and they were treated for 6 weeks. At 12 weeks of age, HSD-fed rats were further subdivided into the following two groups: HSD continued (n = 10) or 0.001% esaxerenone (w/w) was added to the HSD (n = 10). The concentration of esaxerenone in HSD was calculated on the basis of our previous report [42], where DSS rats were gavaged daily with esaxerenone at a dose of 1 mg/kg body weight. In the survival study, all HSD-fed animals died by 24 weeks of age. However, we continued the experiment until 28 weeks of age (16 weeks of intervention), at which point only two animals in the esaxerenone treatment group were still alive. In the cardiac function study, we administered the intervention for 4 weeks and continued the experiments until 16 weeks of age.”
Comment-4
Furthermore, when describing the results regarding the effect of esaxerenone on cardiac fibrosis, the authors note that: “However, esaxerenone treatment in the HSD-fed rats blunted the upregulation of these cardiac fibrotic markers”. However, one sees that in the presence of esaxerenone, the mRNA expression of the fibrotic markers is evidently reduced compared to the HSD-fed rats, but not blunted since the levels remain at least two-fold higher than the ones in LSD-fed rats (Figure 3B-E).
Response
Thank you for pointing out this issue. We agree with your comment that esaxerenone did not completely prevent the upregulation of cardiac fibrotic markers. Therefore, we have revised the text, as follows:
Results (Page 6, lines 162-163)
“However, esaxerenone treatment in the HSD-fed rats reduced these cardiac fibrotic markers significantly.”
Comment-5
In addition, in an effort to establish the effect of esaxerenone on cardiac oxidative stress, the authors have chosen to investigate the mRNA expression levels of gp47 and gp22 phox and perform HNE immunostaning. It might strengthen their study if they looked into lipid peroxidation by performing an MDA assay, or the occurrence of increased protein oxidation. TUNEL assay might also help decipher any beneficial effects of this compound on apoptosis. With such mortality levels (100%) in the HSD fed group, it is necessary to further study the occurrence of apoptosis or other forms of cardiac myocytes cellular death.
Response
Thank you for your excellent suggestions. In accordance with your suggestions, we performed the MDA assay on left ventricular tissue, and we added text to the revised manuscript, as follows:
Results (page 7, lines 193-196)
“Furthermore, LV tissue levels of malondialdehyde (MDA), which is an index of lipid peroxidation, were sharply increased in HSD-fed rats compared with those from LSD-fed DSS rats (Figure 5C). Esaxerenone treatment caused a significant decrease in MDA level in cardiac tissues from DSS rats.”
Discussion (page 10, lines 286-287)
“Moreover, lipid peroxidation, which was evaluated using MDA levels, was greatly increased in HSD-fed DSS rats, but it decreased in esaxerenone-treated DSS rats.”
Methods (page 12, lines 371-376)
“4.10 Cardiac tissue level of MDA
LV tissue homogenates were used to determine the MDA levels using the thiobarbituric acid reactive substances method in accordance with the protocol that was provided with the assay kit (Elabscience, Houston, TX, USA). Protein concentration in the tissue homogenates was measured by the Bradford assay, and MDA levels were normalized to the homogenates total protein content. The data are presented as nmol/mg protein.”
In the present study, we mainly evaluated cardiac fibrosis, inflammation, and oxidative stress. However, on the basis of these suggestions, we are planning to perform an individual future study that will focus on apoptosis and cardiac myocyte death in detail.
Comment-6
Last but not least, one has to report the many misspelling errors in the text.
Since this MS is due to be published in a special issue focusing on the Renin-angiotensin aldosterone system (RAAS), it should at least contain data on the signaling pathways mediating the beneficial effects of the system.
Response
We apologize for the typographical errors. We have corrected these errors and carefully revised the entire manuscript.
Recent studies demonstrated that serum and glucocorticoid-regulated kinase (SGK)-1 plays an important role in angiotensin or aldosterone-induced cardiac fibrosis and remodeling. Therefore, on the basis of the suggestions that were provided, we evaluated SGK-1 mRNA expression and found that this mRNA level increased in HSD-fed DSS rats and decreased in esaxerenone-treated rats. These data suggest that reduced cardiac fibrosis that is observed in DSS rats that were treated with esaxerenone may be due to the reduced SGK-1 mRNA level. We have added this information into the revised manuscript, as follows:
Results (page 6, lines 163-165)
“Moreover, serum and glucocorticoid-regulated kinase (SGK)-1 mRNA expression increased in HSD-fed rats but decreased in esaxerenone-treated rats (Figure 3F).”
Discussion (page 9, lines 259-265)
“Furthermore, activation of the renin–angiotensin system plays an important pathophysiological role in hypertensive cardiac fibrosis and remodeling. Recent studies have demonstrated that SGK-1 plays a critical role in angiotensin or mineralocorticoid-induced cardiac fibrosis [31,32]. In the present study, we found that SGK-1 mRNA expression increased in HSD-fed DSS rats. However, treatment with esaxerenone significantly reduced SGK-1 mRNA levels, which could be associated with the reduction of cardiac fibrosis.”
Methods (page 11, line 358-page 12, line 361)
“The mRNA expression of NPPA, NPPB, MYH7, TGF-β, collagen types I and III, PAI-1, TNF-a, IL-6, CXCL8, gp47phox, p22phox, and SGK-1 was analyzed by real-time PCR using an ABI Prism 7000 system with Power SYBR Green PCR Master Mix (Applied Biosystems, Foster City, CA, USA).”
This manuscript is a resubmission of an earlier submission. The following is a list of the peer review reports and author responses from that submission.
Round 1
Reviewer 1 Report
This is a simple, one setup/one parameter study providing a piece of original knowledge. There are few points which need clarification or correction.
Comments
- A correct expression of SAM is the sympathetic-adrenomedullary (SAM) system.
- Introduction, line 47 – Hyperreactivity cannot be physiological or pathological. “Physiological” means healthy functioning. Hyperreactivity can be hormonal, cardiovascular etc. In case you mean activation of SAM system and HPA axis by stress stimuli, then it is neuroendocrine hyperreactivity.
- I know well that there is a promiscuity in the use of terms stress and stressor in the literature. However, the authors might like to avoid phrases such as “reactivity to stress” or “response to stress” because stress is the response. That should be checked throughout the manuscript.
- In lines 54-56 the authors correctly state that they confirmed earlier research. That sentence should however be rephrased so that at least equal accent is given to earlier studies and to the recent study by the authors. By the way, activation of RAAS by acute stress stimuli is known for many years.
- Was the volume of plasma stored frozen for a long time equal for all samples and all subjects? If so, it would be a good argument to strengthen the validity of the approach, as the volumes can be reduced by long-term freezing.
- There are few important methodological details, which need to be described when measuring aldosterone concentrations. In particular, it should be stated which parts of the test were performed in sitting and which in standing position, going to a quiet room, etc. Postural changes (from sitting to standing) usually performed during psychosocial stress tests are inducing mild elevation of aldosterone release (Mlynarik et al., J. Physiol. Pharmacol. 2007). The authors might like to mention this fact in the Discussion.
- Another methodological detail to be mentioned is stating the time when the intravenous catheter was inserted.
- The sentence starting in line 105 and ending in line 107 does not belong there and should be omitted.
- It is not clear what do you mean by “baseline” aldosterone. Was it before the stress procedure? How long was the subject sitting and how long was it after the venipuncture?
- I am not sure that the first sentence of the Discussion is needed.
- According to above stated comments, also the statement in line 173 should read “…proposed generalized neuroendocrine hyperreactivity to acute stress stimuli in hypertension.”
- The authors may consider mentioning other possible consequences of increased aldosterone concentrations, namely the effects on mood. Although aldosterone was for a long time not expected to influence mental functions, it is nowadays clear that it plays an important negative role in anxiety and depression, see Jezova et al., Curr Protein Pept Sci. 2019.
- Please, do not forget to replace the word “physiological” in the last paragraph of the Discussion and in the Keywords.
Reviewer 2 Report
In their manuscript entitled: “Cardioprotective effects of a nonsteroidal mineralocorticoid receptor blocker, esaxerenone, in Dahl salt-sensitive hypertensive rats”, Asadur Rahman et al. report their results showing the beneficial effects of esaxerenone in Dahl salt-sensitive hypertensive rats.
MAJOR CONCERNS
Given that as the authors themselves report in the Introduction: “Several basic and clinical studies have already demonstrated the tolerability and suitability of esaxerenone in patients with essential hypertension [16,17] or in those with hypertension and type 2 diabetes [18,19]. Although an ongoing clinical study [20] aims to assess the impact of esaxerenone in patients with hypertension and heart failure….” both the novelty and scientific significance of this study consist an issue.
Another major concern is the experimental design of the study. In particular, as the authors state: “HSD-fed DSS rats died by 24 weeks of age (18 weeks of HSD feeding), whereas 100% of the LSD-fed DSS rats were alive even at 28 weeks, suggesting that the HSD-fed rats exhibited a relatively shorter lifespan (Figure 1A). However, 20% of the HSD-fed rats with concomitant intervention of esaxerenone were alive at 28 weeks”. Looking into the materials and methods section one realizes that in fact the 20% corresponds to the 2 rats that survived, out of the 10 that originally belonged to this group. This cannot be accepted as statistically significant. The one group has null survival and the other one, that is supposed to prove the beneficial effect of esaxerenone, contains only 2 rats. How can this be reliable? Unfortunately, this study has to be repeated starting with a much higher number of animals, since the authors intended to further divide them as the study progressed.
Another concern that confuses the reader is the time points that the echocardiographic data and the tissue samples were collected. Since the HSD-fed DSS rats all died, the authors collected data from diverse time points that vary and are not consistent, so that one can extract comparable conclusions between the various experimental groups.
Furthermore, when describing the results regarding the effect of esaxerenone on cardiac fibrosis, the authors note that: “However, esaxerenone treatment in the HSD-fed rats blunted the upregulation of these cardiac fibrotic markers”. However, one sees that in the presence of esaxerenone, the mRNA expression of the fibrotic markers is evidently reduced compared to the HSD-fed rats, but not blunted since the levels remain at least two-fold higher than the ones in LSD-fed rats (Figure 3B-E).
In addition, in an effort to establish the effect of esaxerenone on cardiac oxidative stress, the authors have chosen to investigate the mRNA expression levels of gp47 and gp22 phox and perform HNE immunostaning. It might strengthen their study if they looked into lipid peroxidation by performing an MDA assay, or the occurrence of increased protein oxidation. TUNEL assay might also help decipher any beneficial effects of this compound on apoptosis. With such mortality levels (100%) in the HSD-fed group, it is necessary to further study the occurrence of apoptosis or other forms of cardiac myocytes cellular death.
Last but not least, one has to report the many misspelling errors in the text. Since this MS is due to be published in a special issue focusing on the Renin-angiotensin-aldosterone system (RAAS), it should at least contain data on the signaling pathways mediating the beneficial effects of the system.